# Hazelnut and Walnut Nutshell Features as Emerging Added-Value Byproducts of the Nut Industry: A Review

**DOI:** 10.3390/plants13071034

**Published:** 2024-04-06

**Authors:** Carlos Manterola-Barroso, Daniela Padilla Contreras, Gabrijel Ondrasek, Jelena Horvatinec, Gabriela Gavilán CuiCui, Cristian Meriño-Gergichevich

**Affiliations:** 1Doctoral Program in Science of Natural Resources, Universidad de La Frontera, Temuco 4811230, Chile; carlosignacio.manterola@ufrontera.cl (C.M.-B.); g.gavilan02@ufromail.cl (G.G.C.); 2Scientific and Technological Bioresources Nucleus (BIOREN-UFRO), Universidad de La Frontera, Temuco 4811230, Chile; daniela.padilla@ufrontera.cl; 3Laboratory of Physiology and Plant Nutrition for Fruit Trees, Faculty of Agricultural Sciences and Environment, Universidad de La Frontera, Temuco 4811230, Chile; 4Laboratory of Soil Fertility, Faculty of Agricultural Sciences and Environment, Universidad de La Frontera, Temuco 4811230, Chile; 5Department of Soil Amelioration, Faculty of Agriculture, University of Zagreb, 10000 Zagreb, Croatia; gondrasek@agr.hr (G.O.); jhorvatinec@agr.hr (J.H.); 6Department of Agricultural Production, Faculty of Agricultural Sciences and Environment, Universidad de La Frontera, Temuco 4811230, Chile

**Keywords:** antioxidants, nutshells, physicochemical properties, valued raw material

## Abstract

The hard-shelled seed industry plays an important role in the global agricultural economy. In fact, only considering hazelnut and walnut, the global nut supply is over 5.6 tons. As a result considerable amounts are produced year by year, burnt or discarded as waste, bypassing a potential source of valuable compounds or features. This review deals with the recent scientific literature on their chemical composition as well as functional applications as an approach to sustain the utilization of the main byproduct derived from industry. Indeed, nutshells have received great interest due to their lignin, antioxidant, physical and mechanical features. It was found that these properties vary among cultivars and localities of plantation, influencing physical and structural features. The inconsistencies regarding the above-mentioned properties of nutshells lead to exploring the status of hazelnut and walnut shell applications in sustainable bio-economy chains. In fact, in terms of potential applications, the state of the art links their use to the construction industry and the manufacture of materials, such as resin or plastic composites, particleboards or construction panels, or vital infrastructure and as a filler in cement pavements. However, their current use continues bypassing their great antioxidant potential and their interesting chemical and mechanical features.

## 1. Introduction

Hard-shelled seeds enclosing edible seeds or kernels are referred to as nuts in the agroindustry; some are “biological seeds” or drupes such as almond (*Prunus amygdalus*), brazilian nut (*Bertholletia excelsa*), cashew nut (*Anacardium occidentale*), macadamia nut (*Macadamia integrifolia*), pecan (*Carya illinoinensis*), pistachio (*Pistacia vera*) and walnut (*Juglans regia*), while others are hard-shelled plant fruits or nuts such as chestnut (*Castannea sativa*) and hazelnut (*Corylus avellana*). In production terms, the hard-shelled seed industry plays an important role in the global agroeconomy [1]. Nut production has grown greatly over the past decade, with 4.6 million metric tons (t) in the 2019/2020 season [1]. Almonds were the most produced nuts (31%) in the world, followed by walnuts (21%), cashews (17%) and hazelnuts (12%). Overall, the value of nut production has increased steadily over the past decade at an average rate of USD 1.9 billion per year, reaching USD 35.6 billion for the 2019/2020 season [2,3]. As a result of nut industrialization processes, considerable amounts of byproducts and residues are produced year by year, being stored or burnt as fuel for heaters or discarded as nonvalued waste [4,5,6]. Barbu et al. [7] reported that shell residues derived from walnuts and hazelnuts comprise an amount of 646,818 and 353,807 t, respectively, each year. For the last decade, nutshells have been receiving much increasing interest because of their biochemical performances [8,9], and the nutshells in both species have gained attention as the main byproduct of each productive industry. Considering nutshell weight value (50% and reaching up to 60% *w*/*w*) [8], the utilization of nutshells for the extraction of phenolic compounds has been explored, with the purpose of obtaining natural valued additives (Figure 1). Contini et al. [10] reported interesting concentrations of total phenolic compounds (TPC) in the shells of hazelnut samples (56.6 mg GAE g^−1^ DW) for cultivars Tonda Gentile Romana, Tonda di Giffoni (TDG), Tonda Gentile delle Langhe and Tombul processed at different roasting temperature and times. In addition, Manterola-Barroso et al. [9] reported an oxygen radical absorbance capacity (ORAC) of approximately 2100 μMol TE g^−1^ DW in shell samples of cultivar Tonda di Giffoni. In relation to their chemical composition, ranges of hemicelluloses (25–30%), cellulose (26–34.6%), lignin (40–43%) and extractives (3.3–4%) content have been reported [7,11,12]. Likewise, walnut shells present very similar biochemical features to hazelnut shells. In terms of antioxidants, Queirós et al. [13] reported TPC values of 31.79 mg GAE g^−1^ DW and trolox equivalent antioxidant capacity (TEAC) around 18.86 mg TE g^−1^ DW in shell methanol extracts. Moreover, the chemical composition was described as 46.6% holocellulose, 49.7% polysaccharides, 29.9–49.1% lignin, 25.4% α-cellulose, 10.6%, total extractives and 0.7% ashes [11,12,13]. In this sense, it is important to highlight the potential application and/or use of compounds such as lignin, whose chemical conformation is generated based on phenolic polymers with high antioxidant capacities and which, due to such characteristics, have become very prominent in a wide variety of areas of industrial development [14]. Ultimately, nutshells are used to decrease plain cement dosages in cementitious products, improving mechanical and durability properties. Moreover, hazelnut shells help mitigate the oxidative aging of asphalt binders [15], increasing the reduction in the nut industry’s CO_2_ emissions [15,16,17]. Finally, based on the foregoing background, the aim of this review was to explore the status of potential hazelnut and walnut nutshell applications in circular bio-economy chains.

## 2. Productive Context and Expectative of Hazelnut and Walnut Shells: Potential Byproduct Availability/Accessibility

Hazelnuts are harvested all over the world from approximately one million ha in 38 countries, resulting in a 2020/2021 world production of over 1.1 million t [3]. Turkey is the major producer, with a production of 665,000 t (2020/2021), followed by Italy (140,560 t), the USA (64,410 t), Chile (52,100 t), Azerbaijan (49,465 t) and Georgia (32,700 t) [3,18] (Figure 2). Chile has become the main producer country from the Southern Hemisphere, with an off-season nut supply of approximately 52,100 t (4.73% of total world production) in the 2021/2022 season [18] and an estimated production of 136,000 t year^−1^ for 2030 [18,19]. This nutshell has a mass value of over 50% of the total fruit weight (*w*/*w*) in stabilized nut [9,20]. It is becoming the main hazelnut byproduct production. This means that about 400,000 t are produced worldwide year by year [3,7], and there is a 27,000 t year^−1^ production of nutshells in Chile. These facts support a high nutshell flow and volume production, in addition to their exponential growth over time.

Walnut global production was over 4.5 million t for the 2020/2021 season (Figure 2), experiencing an increase of 23% in the last 10 years [3]. Taking only 50% into account as weight value, and reaching up to 60% nutshell yield (*w*/*w*) in relation to kernel (40–50%) [8], there were approximately 2.2 million t of walnut byproduct (nutshell) produced. Walnuts are harvested all over the world, wherein China is the major producer, providing 34.6% of the total world walnut supply, followed by the United States (13.3%), Iran (8.94%), Turkey (4.69%), Mexico (3.48%) and Chile (1.83%) [3] (Figure 2). In fact, and contextualizing the Chilean national market, walnut exports reached 136,000 t for the 2022 season [8]. As in the case of hazelnuts, walnut byproduct production generates a huge volume of nutshells with an exponential industrial potential over time. In fact, it is one of the most widely produced nuts globally. In this sense, and articulating the state of the art from a commercial and industrial point of view, there is a great opportunity to reincorporate these byproducts in the industrial chain due to the wide and interesting range of intrinsic properties and characteristics that they exhibit, ranging from chemical and biochemical to structural and physical.

## 3. Nutshell as Source of Valuable Biochemical Components

### 3.1. Main Chemical Composition

Hazelnut shells are structurally composed of hemicelluloses (25–30%), cellulose (26–32%), lignin (40–43%) and other H_2_O soluble extractives (3.3–4%) [6,7]. Other authors reported for some commercial hazelnut nutshell samples a lignin content of 36% w/wd with an extraction yield of 25% wd/ws (ws: weight of starting material and wd: weight) [14]. Likewise, Husainie et al. [21] reported 30.2% “Klason” lignin (Klason method by acid hidrolysis), 28.9% cellulose and 11.3% hemicellulose for commercial hazelnut. However, other authors have reported (by Kurschner–Hoffner method and acid hydrolysis) a lignin content of 46.70% weight percent (wt%), 26.86 wt% of cellulose and 21.65 wt% of hemicellulose [22]. On the other hand, walnut shell chemical composition is very similar to other wood biomass or nut-industry byproducts such as hazelnut shells, because cellulose, hemicellulose and lignin are the main components. In fact, walnut shells are constituted by 49.7% polysaccharides, 29.9% lignin and 10.6% other H_2_O extractives [13]. Moreover, Domingos et al. [23] reported 35% lignin and 55.2% holocellulose (30.4% and 24.9% α-cellulose and hemicelluloses, respectively), whereas Wei et al. [24] reported a walnut lignin content of 53.5%, hemicelluloses of 21.32% and cellulose of 25.16%. This is quite similar to the values reported by Jovicic et al. [25], a study in which the authors reported values of cellulose (32.62% ± 0.22), lignin (53.87% ± 0.85), holocellulose (42.45% ± 0.78) and extractives (2.46% ± 0.80) for walnut shell samples (cultivar Šejnovo) (Table 1). Thus, all modern methods for the determination of soluble lignin are based on the hydrolysis and solubilization of carbohydrates with different types of acids (mainly with 72% H_2_SO_4_). Holocellulose corresponds to the H_2_O-insoluble fraction of carbohydrates just separated before reactions with CH_3_ COOH and NaClO_2_ (25%). Finally, regarding α-cellulose, it corresponds to the fraction that remains insoluble; thus, it can be separated after several washes with NaOH.

### 3.2. Antioxidant Properties

Regarding the hazelnut antioxidant features, Manterola-Barroso et al. [9] reported a mean ORAC antioxidant capacity (AC) of 2119 μmol TE g^−1^ DW in control treatments of TDG shell samples, demonstrating the interesting antioxidant potential of hazelnut shells. Moreover, Pelvan et al. [32] have reported AC (ORAC method) values from roasted hazelnut skin samples, which could be a part of and an approach to the antioxidant capacity of the hazelnut shell of approximately 1219 μmol TE g^−1^, also TPC content (Folin-Ciocalteu) of 72.25 mg GAE g^−1^ and RSA (DPPH method) of 1.01 mg mL^−1^ (extract). Other authors also reported total phenolic compound (TPC) values of around 180 μg GAE g^−1^ DW and a radical scavenging activity (RSA) of 1110 μg TE g^−1^ DW in samples (methanolic extracts). However, for hazelnuts, Masullo et al. [26] reported a higher value of TPC (340 μg GAE mg^−1^) for methanolic (70%) shell extracts (cultivar Nocciola di Giffoni). Likewise, Di Michele et al. [27] reported ranges between 3.5 to 12 mg GAE g^−1^ from 22 different extract preparation and characterization of hazelnut (Tonda Gentile Romana) shells. These findings are concomitant with the ranges of TPC reported by Yuan et al. [28] in hazelnut cultivar Lewis shell samples (6–7.7 mg GAE g^−1^). In relation to the studies above on the quantification and characterization of antioxidant capacity and phenolic compounds present in nutshells, they are directly related to the production of high-quality antioxidants for different purposes (mainly for pharmaceutical and cosmetic industries), pointing directly to a natural source of phenolic compounds, possibly meaning a period of characterization and behavioral determinations of antioxidant and phenolic compounds. Therefore, there is currently no direct application of walnut shells in relation to their antioxidant qualities.

Regarding walnut shell antioxidant properties, Queirós et al. [13] reported TPC values of approximately 31.79 mg GAE g^−1^ DW from commercial Portuguese walnut shell samples, but the authors did not indicate the genotypic description and location. Complementarily, the same authors stated RSA values of around 18.86 mg TE g^−1^ DW, demonstrating that walnuts represent higher antioxidant properties in comparison with other nut species such as almond and pine nut, focused on nutshell valorization as byproduct. Regarding the state of the art, the skin and shell of walnuts, which are a rich sources of phenolics, are responsible for effective scavenging of free radicals [28,29,33]. Moreover, regarding phenolics, Han et al. [34] reported a mean of 20.6 mg GAE g DW^−1^ for commercial nutshell samples from California (United States). From the north of Spain (Basque country), Herrera et al. [30] quantified and qualified the phenolic compounds by gas chromatography (GC) and GC-mass spectroscopy (GC-MS), identifying groups of phenols, phenolic acids and lignin units (2.22 mg g^−1^), lignans (0.30 mg g^−1^), stilbenes (0.02 mg g^−1^), flavonoids (0.69 mg g^−1^) and unknown fractions (0.23 mg g^−1^). Therefore, by relating the qualities of extracts and pure phenolic compounds from the shells of *J. regia* to their current applications, they might be used as natural antioxidants and alternatives to synthetic antioxidants, such as BHT (2,6-ditert-butyl-4-methylphenol) [31].

However, the authors analyzed a walnut assortment provided by local growers but did not mention the origin of the studied vegetal material. All the previous authors’ findings were based on methanolic extracts of hazelnut and walnut shell samples from several origins. Moreover, it is important to highlight that the inconsistencies in antioxidant determinations (RSA, ORAC and TPC) among the authors could be attributed to environmental factors, as Manterola-Barroso et al. [9] reported, including weather and plantation conditions, as well as the evaluated genotype. Regarding both nutshell species’ antioxidants features, there was not enough information, probably due to their main application and research area, which is closely related to the ash production as raw material for amendment by pyrolysis. However, it is important to highlight that the inconsistencies in antioxidant determinations, such RSA, ORAC and TPC (Table 1), among the authors could be attributed to the hemisphere, latitude and locality of evaluated samples, similarly to reports by Manterola-Barroso et al. [9] for hazelnuts. Likewise, in relation to the main chemical composition, the ranges varied significantly less and all values were similar, values that probably match due to homogeneity in the determination methodologies employed.

Finally, in terms of potential applications, the vanguard aims their use toward the construction industry and the manufacture of materials, such as resin or plastic composites and particleboards or construction panels [7,35,36,37]. Sandoval et al. [15] emphasized their use on the mitigation of oxidative aging in asphalt layers, as a filler in cement pavements. However, their current use continues to be as a fuel and material of low added value [5,6], ignoring their great antioxidant potential (phenolics) and their interesting chemical composition.

## 4. Physical Features of Hazelnut and Walnut Shells

### 4.1. Density and Solubility

In relation to physical terms, hazelnut shells’ real density and theoretical density were determined by Barczewski et al. [37]. The authors reported values for hazelnut and walnut shell samples (unknown cultivars obtained from local growers in Poland) that round from 1.168 to 1.19 g cm^−3^ and 1.155 to 1.211 g cm^−3^, respectively. Likewise, Matin et al. [38,39] recently reported shell density values of 1.05 ± 0.09 and 1.3 ± 0.21 g cm^−3^ for hazelnut cultivars Istarski duguljasti and Rimski okrugli, respectively. In the case of walnut shell samples, Barczewski et al. [37] determined values between 1.098 and 1.164 g cm^−3^ for real density and 1.103 and 1.207 g cm^−3^ for theoretical density (Table 2). In fact, all authors showed results determined on different volumes of milled byproducts.

Finally, regarding solubility, Pirayesh et al. [11] reported solubility data for hazelnut samples in alcohol benzene (2/1) of 2.0%, whereas for walnut shells they were 3.2%. The same author showed determination with NaOH (1%) of 50.4 and 35.2% for hazelnut and walnut, respectively. In addition, hot- and cold-water solubility ranged from 18.2 to 20.9% for hazelnut and between 7.6 and 10.2% for walnut shell analyzed samples.

### 4.2. Mass Yield and Morphological Features

As mentioned above, both nutshells are the main byproducts of the nut industry, and both represent a weight percentage (in relation to the total weight of the whole nut) between 50 and 60%, respectively (*w*/*w*) [9]. Likewise, Milošević and Milošević [40] reported results (from 12 different cv of hazelnuts) on shape, informing values of 20.85 mm length, 19.93 mm width and 1.73 mm for shell thickness. Regarding walnut shells, the kernel/shell relation is quite higher than in hazelnuts. In fact, the total weight of the nutshell ranges between 50 to 62% (*w*/*w*) of the total nut biomass weight [8,41,42,43] (Figure 3). On the other hand, Angmo et al. [42] reported walnut shell thickness mean values of 3.80 mm, with 53.40 mm for nut length, 48.00 mm for nut diameter and 62.68% of nutshell yield for Cv Skara Nurla Temisgam and Dhomkhar. However, Ozkan and Koyuncu [42] reported lower values of shell thickness (1.13 mm), nut length (35.28 mm) and diameter (29.39 mm) means for 10 Turkish walnut genotypes. Therefore, both byproducts exhibit large hardness and toughness [5,6]. In relation to hazelnut shell thickness of around 1.75 mm [9,20], walnut shell thickness ranges between 1.1 to 1.7 mm according to reports by Koyuncu et al. [43] (Table 2).

**Table 2 plants-13-01034-t002:** Average physical features of hazelnut and walnut shells.

Nutshell Species	Density	Solubility		Morphology (nut traits)			References
		NaOH	H_2_O	L	W	STh	
	(g cm^−3^)	(%)		(mm)			
Hazelnut	1.0–1.3	50–50.4	18–20	18.8–25.9	17.2–22.7	1.3–1.7	[9,11,20,37,38]
Walnut	1.1–1.2	35.2	7–10	38.2–50.4	29.2–34.4	1.1–3.8	[11,37]

### 4.3. Nutshell Cracking Point

Structurally, nutshells (rich in lignocellulosic compounds) are characterized by exhibiting high hardness and firmness [5,6,44]. Valentini et al. [45] studied such properties, reporting values of force required for penetration of shells (from 18 hazelnut cultivars from different countries in Europe: Germany, France, Italy, Spain and England). These values ranged from 48.0 ± 3.8 to 185.7 ± 4.0 Newton (N) (an average of 87.89 ± 4.0). In addition, this confirms a strong correlation (r = 0.94) between the thickness of the shell (6% max humidity) and the force required for its penetration (N). In this sense, Kabas et al. [45] affirmed the values above, reporting these “crack points” for hazelnut shells (kernel in shell) at 138.05, 55.73 and 89.23 N for longitudinal, transverse and suture orientations, respectively. Otherwise, there are published values of the ranges [46] where the average of hardness determination of 10 Turkish hazelnut cultivars shell samples were 294 N (ranging between 191.24 ± 32.70 and 519.56 ± 52.24 N). In relation to walnut nutshells, Koyuncu et al. [43] reported values of 333.0 (longitudinal), 472.0 (transverse) and 441.0 N (suture) for shell samples (2001–2002 season) of Yalova-3 cultivar. Likewise, Sharifian and Derafshi [47] determined values of force to cause nutshell fracture in West Azerbaijan (Iran) walnut nutshell samples (2006 season) of 270.40 N L (longitudinal), 499.20 N T (transverse) and 424.70 N S (suture) (Table 3). There are other authors that have studied the effect of different types of compression (flat, conical and spherical) and the interaction and relationships between force (N) and deformation (mm) of walnut shell (in-shell) samples; they determined forces to crack point of 211.83 N for spherical compression, 328.55 N for conical compression and 176.84 N for flat compression [48]. However, the ranges observed were possible due to several factors, such as species, evaluated cultivar, locality of plantation, soil–climate parameters and the point or direction where the force was generated. But, in all reports, the basis was to determinate the maximum force to fracture the nutshell in different positions (of the nut) with different methods or in relation to the food industry (point of cracking in-shell kernels).

## 5. Industrial Reincorporation for Potential Applications

Agricultural and forestry byproducts have several benefits, among which are reduced costs, availability, biodegradability, renewability and increased flexibility [19,50,51,52]. Only a few scientific attempts have been made to solve the utilization of these byproducts from a circular bio-economy point of view [52]. In fact, findings reported by Balart et al. [35,36] indicate that hazelnut shells can be optimally used to reinforce filler in fully biodegradable composites with a polylactic acid matrix (PLA). Moreover, hazelnut shell samples in combination with jute (*Corchorus capsularis*) fiber have been reported as a natural and biodegradable filler in asbestos-free nonmetallic organic friction composites in which phenolic resin was used as a binder. Following the same industrial area, walnut, hazelnut and sunflower (*Helianthus annuus*) husks have been used as fillers for the production of inexpensive, epoxy-based composites, improving certain critical parameters in the evaluated models [37]. As a construction option, Barbu et al. [7] evaluated the influence of hazelnut and walnut shells 10% bonded with melamine urea formaldehyde (MUF) and polyurethane (PUR) on the mechanical and physical properties of particleboards (PBs), demonstrating the suitable use of nutshells for improving the performance of dimensional stability and “Brinell hardness” (determined by the penetration of an object into the studied material) in a PB model [7]. All this with a view to the production of natural fiber composites (NFC) or boards, which is the most dynamic development direction according to the technology in composites and particleboard, as it has been growing for the past years.

On the other hand, concerning the environmental and functional implications of byproducts such as hazelnut and walnut shells, their richness in phenolic compounds and antioxidant properties have been scarcely studied regarding their potential application with industrial purpose within a circular bio-economy and sustainable context [53,54,55,56]. These materials may be considered as proper to reduce the oxidative aging effect on materials involved in roads infrastructure. In relation to cementitious pavements, nutshells are often used to decrease plain cement dosages in concrete pavements, improving the mechanical and durability properties and mitigating the oxidative aging of concrete and asphalt layers, respectively, which also helps in the reduction in these byproduct industries’ CO_2_ emissions and footprints [16,17]. In fact, Sandoval et al. [15] determined that agroindustrial wastes derived from grape pomace (*Vitis vinifera*), with interesting antioxidant potential, mitigated the oxidative aging on asphalt on an experimental scale, a process that contributes to all failure factors in durability, permeability and mechanical resistance. Nowadays, scientific initiatives have emerged after an association between academy and private companies has been conducted to increase the valorization of these byproducts as an antioxidant modifier for asphalt binders conferring resistance and protection of the aging processes in the asphalt matrix. Based on the above, ANID (Fondef) VIU20P0027 “Modified eco-asphalt with an antioxidant additive obtained from hazelnut shells as the main waste product of the industry”, has obtained interesting results in terms of the mitigation of oxidative aging in asphalt binders (around 25%), using an antioxidant additive made from hazelnut shells (unpublished data).

Finally, in relation to the foregoing points, it is possible to understand more accurately where the initiatives of a circular bio-economy based on the use or valorization of these byproducts for their application in industrial processes are being directed. However, there are still several gaps and inconsistencies regarding the main properties of both shells, and it is necessary to study and research with more emphasis the interesting properties that constitute these byproducts and thus shorten the time to breaching their potential applications. This is the case since the continuous development of a modern society in connection with technology leads to a high demand for inexpensive ready-to-use products.

## 6. Outlooks

Chemical, antioxidant and physical features of hazelnut and walnut shells were exposed and characterized in view of their valorization as the main byproducts of nut industries. Both nutshells were predominantly composed of lignin (27–52%), hemicellulose (25–30%), cellulose (26–34%) and extractives (3–3.4%). In association with the huge antioxidant potential, there is an argument to be made for an interesting industrial application potential of these byproducts. But, there is another gap that is actually linked to the identification of these compounds (mostly phenolics); so, it would be highly interesting to research in a deeper way about these antioxidants properties, especially the phenolic components influenced by environmental biotic and abiotics factors. Moreover, there are interesting opportunities in the research on the physical properties of both byproducts, such as photocolorimetric features or hardness/cracking points. In this sense, it is important to highlight the potential applications and/or uses of compounds such as lignin, whose chemical conformation is based on phenolic polymers with high antioxidant capacities and which, due to such characteristics, are finding more and more applications in a wide variety of areas of industrial development.

In relation to all the variations reviewed above, there are several factors that may influence physical and structural features (that in some cases are closely correlated to chemical and antioxidant ones), such as orchards location (soil, weather conditions and agronomical management), differences among cultivars or ecotypes and postharvest processes, as Akbari et al. [52] mentioned in their study. Moreover, there are scarce reports of hazelnut and walnut physical, mechanical and colorimetric properties, ORAC antioxidant capacity (the method for which there is limited reports on for nutshells) or other features that could be useful for several industrial potential applications, such as development of alternative colorants, fillers in construction materials, asphalt and cementitious infrastructure sciences aims. Based on the background information reviewed and consulted, it can be inferred, due to the lack of clarity of the authors in their description of the vegetal material (nuts) used in their experiments and studies, that a possible effect of the great gap between the representative nut industry and science, innovation and development is a scarcity of precision at the time of describing chemical and biochemical properties, as well as physical and mechanical ones. Finally, analyzing all previously exposed, the detected gaps should be the subject of further research in a transitional context from nutshells as waste to added-value materials in a bioeconomy context. Therefore, it is imperative to investigate more deeply and look forward with the aim of generating the basis for the development of the quality parameters of these byproducts from nut industry.

## Figures and Tables

**Figure 1 plants-13-01034-f001:**
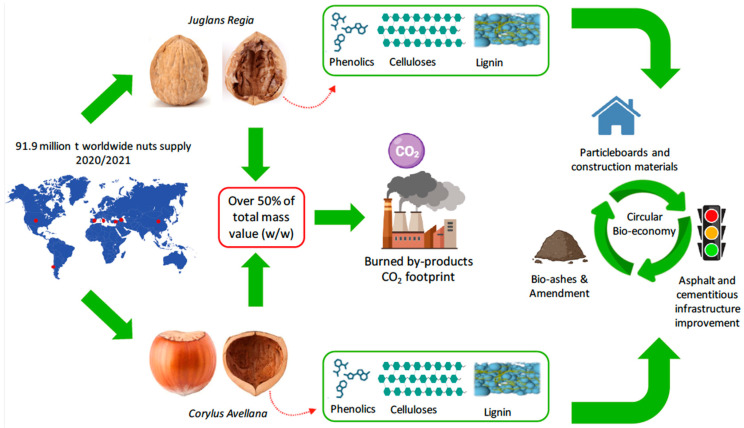
Illustration of the production process, utilization and potential applications of hazelnut and walnut shells.

**Figure 2 plants-13-01034-f002:**
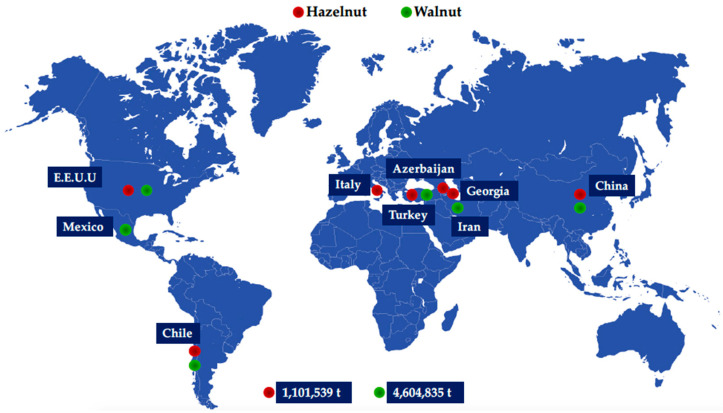
Global production of hazelnut and walnut fruits (FAOSTAT, 2023; ODEPA-CIREN, 2021).

**Figure 3 plants-13-01034-f003:**
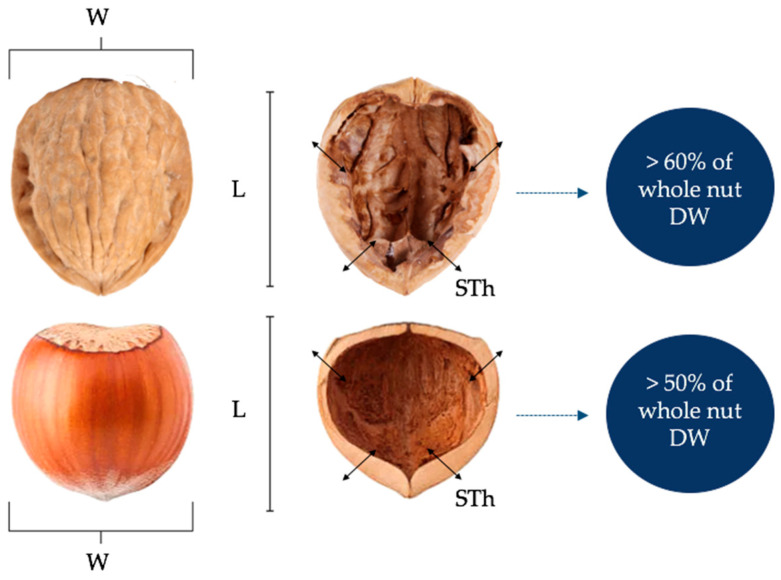
Mean mass yield and morphology of whole nut and nutshell of hazelnut and walnut. W: width; L: length; STh: shell thickness; DW: dry weight.

**Table 1 plants-13-01034-t001:** Main biochemical features of hazelnut and walnut shells based on the following reported information. Values and ranges were interpreted as means. RSA (radical scavenging activity), TPC (total phenolic compounds) and AC (ORAC antioxidant activity).

Nutshell Species	Lignin	Hemicellulose	Cellulose	Extractives	RSA	TPC	AC	References
	(%)				(μg TE g^−1^ DW)	(mg GAE g^−1^ DW)	(μmol TE g^−1^ DW)	
Hazelnut	36–46	21–30	26–34	3–3.4	-	56.6	-	[7,9,10,11,14,22,26,27,28,29]
					1110	0.18	2119	
					-	0.34	-	
					-	3.5–12	-	
					-	72.25	-	
					1.01 mg mL^−1^	-	1219	
Walnut	29–53	21–24.9	25–32	4–10.6	18,860	31.79	-	[13,23,24,25,30,31]
					3.14–7.17 μg mL^−1^	3.49 μg g^−1^	-	

**Table 3 plants-13-01034-t003:** Hazelnut and walnut hardness properties based on cracking point.

Nutshell Species	Cracking Point			References
	L	T	S	
	(Newton)			
Hazelnut	138–140	55	89	[46,47]
Walnut	270–333	472–499	424–441	[48,49]

## Data Availability

Data sharing not applicable.

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
