# Peer review of "Hazelnut and Walnut Nutshell Features as Emerging Added-Value Byproducts of the Nut Industry: A Review"

_plants, 2024, doi:10.3390/plants13071034_

Round 1
Reviewer 1 Report
Comments and Suggestions for Authors
plants-2639402-peer-review-v1
Hazelnut and walnut nutshell features, as emerging added value by-products of the industry: a review.
Running title: Nutshell features into circular bio-economy.
Carlos Manterola-Barroso, Daniela Padilla, Gabrijel Ondrasek, Jelena Horvatinec, Gabriela Gavilán-Cui-Cui, and Cristian Meriño-Gergichevich
Dear Editor,
The Authors offer a review work of valuable side product hazelnut and walnut shells. They study their applications into sustainable circular bio-economy chains. This review deals with the recent scientific literature on their chemical composition as well as their functional applications as an approach to sustain the utilization of the main by-product derived from the nut-industry.
Comments
-The Authors can describe the general summarize of experimental procedures of the extracting lignin, hemicellulose, etc.
-Antioxidants applications such as active packaging materials should be described with literatures, and should be mentioned about tannic acid.
2. Productive context and expectative of hazelnut and walnut shells: Potential by- 81
product availability/accessibility
-The statistical information could also be given in Tables as well.
-Please describe “klason lignin”.
4.2. Mass yield and morphological features-needs pictures of hazelnut and walnut.

Author Response
Dear reviewer, thanks for comment and suggestions
The Authors can describe the general summarize of experimental procedures of the extracting lignin, hemicellulose, etc.
These procedures were generally explained to allows a basic understand of extraction and determination processes
Antioxidants applications such as active packaging materials should be described with literatures, and should be mentioned about tannic acid
Yes, although our focus was to reveal information about these by-product on construction of self-improving material due their richness in antioxidant, your suggestion appear as very interesting and under this line we are preparing other document with thses potential application.
The statistical information could also be given in Tables as well.
Thanks, that information was described in main text.
Please describe “klason lignin”.
Yes, In text you can find a detailed description (green marked)
Mass yield and morphological features-needs pictures of hazelnut and walnut.
Thanks, Figure with this information was added.
Reviewer 2 Report
Comments and Suggestions for Authors
Nice, concise and informative review covering the important issue of using the by-products of the hard-shelled nuts industry. In my opinion the article as such is on good scientific level and ready to be published. However, given the orientation on the reduction of the waste, I would suggest mentioning also the possibilities to use the other specific by-product of walnuts: seed cover of Juglans regia, which is well-know for its medicinal properties in ethnomedicine and even sold in the health shops in some countries (eg Canada). Seed shell of Juglans regia has been recently studied also for its chemical composition:
https://pubmed.ncbi.nlm.nih.gov/22860453/
https://pubmed.ncbi.nlm.nih.gov/36537558/
Author Response
Dear reviewer
Thanks for suggestion. In our short review several topics could be included, however, we are preparing another document including this important recommendation. This new documnet will be referred to other use and valorization of these by-products.
Reviewer 3 Report
Comments and Suggestions for Authors
Dear authors,
The manuscript entitled (Hazelnut and walnut nutshell features, as emerging added value by-products of the nut industry: a review) is well written and presented, some comments should be taken into consideration before acceptance.
The abstract should be improved.
Line 45, 2017 should be removed.
Line 56, 2022 should be removed.
Line 62, 2019 should be removed.
In Figure1, Graphical abstract should be swapped with illustration.
in figure2, add the names of countries on the map.
Table 2 is difficult to be understand.
In all references remove the year. Also, the guidelines of MDPI for references should be taken into account.
The outlooks and conclusion should be improved.
The English quality should be revised by an expert to improve its performance.
Comments on the Quality of English Language
The English should be revised by an expert or native language speaker.
Author Response
Dear Reviewer, thanks for comments.
-Abstract was improved
-All information about year of plublication was removed from references and adapted to plants editorial guidelines.
Table 2 was improved in a format more comprehensive
Figure 2, name of countries was added.
Outlooks section was improved
English was improved
Round 2
Reviewer 1 Report
Comments and Suggestions for Authors
The Authors mostly well revised the manuscript according to the comments. It can be acceptable for publication in the revised form-present form.
kind regards